# Lipid Peroxidation of the Docosahexaenoic Acid/Arachidonic Acid Ratio Relating to the Social Behaviors of Individuals with Autism Spectrum Disorder: The Relationship with Ferroptosis

**DOI:** 10.3390/ijms241914796

**Published:** 2023-09-30

**Authors:** Kunio Yui, George Imataka, Tadashi Shiohama

**Affiliations:** 1Department of Pediatrics, Graduate School of Medicine, Chiba University, Chiba 260-8677, Japan; asuha_hare@chiba-u.jp; 2Department of Pediatrics, Dokkyo Medical University, Mibu 321-0293, Japan; geo@dokkyomed.ac.jp

**Keywords:** autistic social behavior, arachidonic acid, docosahexaenoic acid ferroptosis, lipid peroxidation, malondialdehyde-modified low-density lipoprotein

## Abstract

Polyunsaturated fatty acids (PUFAs) undergo lipid peroxidation and conversion into malondialdehyde (MDA). MDA reacts with acetaldehyde to form malondialdehyde-modified low-density lipoprotein (MDA-LDL). We studied unsettled issues in the association between MDA-LDL and the pathophysiology of ASD in 18 individuals with autism spectrum disorders (ASD) and eight age-matched controls. Social behaviors were assessed using the social responsiveness scale (SRS). To overcome the problem of using small samples, adaptive Lasso was used to enhance the interpretability accuracy, and a coefficient of variation was used for variable selections. Plasma levels of the MDA-LDL levels (91.00 ± 16.70 vs. 74.50 ± 18.88) and the DHA/arachidonic acid (ARA) ratio (0.57 ± 0.16 vs. 0.37 ± 0.07) were significantly higher and the superoxide dismutase levels were significantly lower in the ASD group than those in the control group. Total SRS scores in the ASD group were significantly higher than those in the control group. The unbeneficial DHA/ARA ratio induced ferroptosis via lipid peroxidation. Multiple linear regression analysis and adaptive Lasso revealed an association of the DHA/ARA ratio with total SRS scores and increased MDA-LDL levels in plasma, resulting in neuronal deficiencies. This unbeneficial DHA/ARA-ratio-induced ferroptosis contributes to autistic social behaviors and is available for therapy.

## 1. Introduction

Lipid peroxidation is a process under which oxidants attack lipids containing carbon–carbon double bonds such as polyunsaturated fatty acids (PUFAs). Lipid pe-roxidation causes cell damage and various human health issues, such as mitochondrial dysfunction in neurodegenerative disorders [1]. PUFA-containing phospholipids are highly susceptible to lipid peroxidation under oxidative stress [2]. The cell membrane includes higher levels of PUFAs, and thus, lipid peroxidation easily occurs under conditions in which reactive oxygen species (ROS) readily react with vulnerable lipids in cell membranes, inducing lipid peroxidation [3]. Lipid peroxidation is metabolized into malondialdehyde (MDA) and 4-hydroxynonenal (4-HNE) [4]. Therefore, oxidative stress has been implicated as an important mechanism in lipid peroxidation in humans, which is linked to MDA and 4-HNE [5]. Lipid peroxidation plays an important pathophysiological role in autism spectrum disorder (ASD) as an abnormal form of lipid metabolism [5]. The peroxidation of PUFAs has emerged as a key driver of oxidative damage to cellular membranes, lead-ing to cell death, such as ferroptosis [6,7]. Ferroptosis is a form of regulated cell death dependent on reactive oxygen species (ROS) and characterized by the accumulation of lipid peroxides [8].

PUFAs play important roles in ferroptosis because of the formation of peroxyl radicals, leading to irreparable membrane damage and cell death [6]. The misregulation or depletion of PUFA-protective enzymes and molecules leads to ferroptotic damage in lipid peroxidation [7]. As 16 ferroptosis-related genes in ASD patients were distinct from those in normal samples, they suggested the critical role of ferroptosis in the pathogenesis of ASD. [9]. Importantly, the presence of ferroptosis by lipid peroxidation in this study was based on proven increased plasma levels of MDA-LDL [10,11]. Ferroptosis is mediated by PUFA lipid peroxidation by condensed mitochondrial membrane densities and smaller volumes than normal mitochondria as well as by the diminished mitochondria crista [11]. DHA triggers ferroptosis via the upregulation of methylenetetrahydrofolate dehydrogenase 2 (MTHFD2) expression [12]. The activity of elongation of very long-chain fatty acid protein 5 (ELOVL5) and fatty acid desaturase 1 (FADS1) lead to ferroptosis sensitization [13]. Furthermore, Arachidonic acid (ARA) facilitates ferroptosis by elongating very long-chain fatty acid protein 5 (ELOVL5) and fatty acid desaturase-1 (FADS1) [13]. Arachidonic acid 15-lipoxygenase promotes ferroptosis by converting intracellular unsaturated lipids into oxidized lipid intermediates and is an important ferroptosis target [14]. It is important to note that ferroptosis-related cell death may be related to neuronal deficiency, resulting in a lipid peroxidation related important factor in the development of ASD. However, there are few studies on the role of ferroptosis.

Regarding the lipid peroxidation product MDA in ASD, previous studies revealed significantly higher plasma MDA levels in 20 children with ASD compared with 20 age-matched controls [15] and that blood MDA levels in 45 autistic children of ages 3–11 years old were higher than those of 42 age-matched controls [16]. However, which variable of PUFA and MDA is the main contributor to the development of autistic social symptoms remains unclear. Oxidative-stress-triggered lipid peroxidation is closely related to a major protein superoxide dismutase (SOD) [17]. SOD controls a variety of reactive oxygen species (ROS) and reactive nitrogen species, controlling broad aspects of cellular life [18] (Wang). SOD related increased oxidative stress altered activities of erythrocyte free radical scavenging enzymes in autism [19]. The SOD enzyme plays an important role in protecting tissues by scavenging oxidative-stress-related reactive oxygen species (ROS) [19].

Previous studies have revealed that children with ASD are more vulnerable to oxidative stress in the form of increased lipid peroxidation and deficient antioxidant defense mechanisms, for example, showing blood cell SOD were significantly higher in 30 autistic children aged 3–15 years as compared with those in 30 healthy children [20]. In another study, MDA levels and SOD activity in plasma were significantly higher in 52 patients aged 3–6 years old with ASD in comparison with 48 age- and gender-matched healthy controls [21]. This finding indicates that children with ASD have higher oxidative stress, suggesting that increased vulnerability to oxidative stress may contribute to the development of ASD [21]. Lipid-peroxidation-related ferroptosis produces increased MDA levels and decreased SOD levels [22,23] via oxidant damage and neuroinflammation [24] and further induces iron metabolism (for example, through the transferrin receptor) with intense oxidative stress and inflammation, in addition to the noted association between MDA and SOD levels [24].

ASD-model mice injected with VPA exhibited neurobehavioral deficits typical of ASD, showing changes in SOD activity and increased lipid peroxidation [25]. However, the association between definitive blood SOD levels and lipid peroxidation in children with ASD remains unclear.

One of the components of the lipid profile, low-density lipoprotein (LDL) plays an important role in brain development. A previous clinical study reported significantly higher blood LDL levels in 22 adult subjects with Asperger’s syndrome (mean age, 40.8 ± 10.8 years) in comparison with 22 age-matched controls (44.6 ± 14.8 years), indicating abnormal cholesterol metabolism [26] (Dzi). Thus, blood LDL levels may contribute to the pathophysiology of ASD.

MDA easily reacts with acetaldehyde to form malondialdehyde-modified low-density lipoprotein (MDA-LDL) [27,28]. MDA-LDL is a good marker of oxidative stress and lipid peroxidation [28]. However, the role of MDA-LDL in the pathophysiology of ASD has not been investigated. This is the first study to examine the role of MDA-LDA in ASD.

Taken together, the following avenues of investigation were proposed: (a) the association between PUFA induced ferroptosis and autistic social behaviors; (b) which type variables of PUFA (omega-3 or omega-6) mainly contribute to MDA-LDL formation; (c) which variable of PUFA and MDA-LDL mainly contributes to the development of autistic social symptoms; and (d) whether the association between lipid peroxidation and the antioxidant SOD may contribute to autistic social impairments.

The outcome-adaptive lasso for selecting appropriate covariates for inclusion in propensity score models accounts for confounding bias and maintaining statistical efficiency [29]. We employed the adaptive Lasso technique to ensure the accuracy of the prediction. Simulations findings and real data examples indicate the better performance of the proposed penalized [30]. The adaptive Lasso is a popular statistical approach based on weighted norm penalties in weights derived from an initial estimate of the model parameter [31], and the adaptive lasso welcomes the variables’ effects in penalty, meanwhile specifying adaptive weights to penalize coefficients in a different manner [31]. 

## 2. Results

### 2.1. Characteristics of the Individuals with ASD

The results of the Mann–Whitney U test revealed that the ages of the ASD and control groups did not differ to a statistically significant extent (U = 63.00, *p* = 0.664). Eighteen individuals were characterized by impaired social communication behaviors (*n* = 18). Their mean total SRS scores were 83.39 ± 35.44 (Table 1). A former clinical study reported that the total SRS score of 29 young men (aged 7–21 years old) with moderate-to-severe ASD was 120.21 [32]: the total SRS scores in the present study were reasonably defined as mild–moderate impairments of social communication behaviors.

### 2.2. Dietary Nutrients

There were no significant differences between the ASD and control groups in weight; height; energy intake; or in the intake of protein (*p* = 0.91), cholesterol (*p* = 0.55), vitamin B2 (*p* = 0.97), vitamin B12 (*p* = 0.97), vitamin C (*p* = 0.18), omega-6 (*p* = 0.44), omega-3 (*p* = 0.50), iron (*p* = 0.55), ans copper (*p* = 0.55) (Table 2).

### 2.3. Plasma Levels of MDA-LDL, SOD and PUFAS

In the ASD group, plasma MDA-LDL levels were significantly higher and plasma SOD levels were significantly lower in comparison with the control group. The plasma DHA level and the plasma DHA/ARA ratio were significantly higher, and the plasma level of adrenic acid (AdA), an omega-6 PUFA, was significantly lower than in the control group (Table 1, Figure 1, Figure 2 and Figure 3).

### 2.4. Gender Difference

Although cognitive functioning is similar across males and females with ASD, males with ASD have a higher perceptual reasoning ability, and females with ASD tend to have more comorbid intellectual impairment than males and require additional support of intervention. Especially, low IQs in the ASD children are more likely to exhibit sex differences in their core symptoms than children with high IQs. Intelligence has an important key role in sex-based differences in the core symptoms of ASD [33]. As shown in Table 5, there were no significant differences in the plasma variables and the total scores of SRS. However, plasma DHA levels tended to increase in 12 male individuals as compared with those in 6 female individuals.

### 2.5. Results of the Multiple Linear Regression Analysis

A multiple linear regression analysis revealed that the plasma DHA level (R^2^ = 0.997, *p* < 0.01) and plasma DHA/ARA ratio (R^2^ = 0.972, *p* < 0.01) were significantly associated with adjustments in plasma variables and the total SRS scores in the two subject groups (Table 3). These findings revealed that the plasma DHA level and the plasma DHA/ARA ratio can predict all the variables of the two groups. The use of plasma alpha-linolenic acid levels as a dependent variable showed the significant contribution of the plasma DHA/ARA level (unstandardized coefficients, *B* = −0.267 ± 0.102, *β* = −0.316, *p* = 0.04) (Table 3). Therefore, the plasma DHA level and plasma DHA/ARA ratio distinguished the ASD group from the control group.

### 2.6. Results of the Adaptive Lasso Analysis

To identify the most effective variables to interpret the sample data, the adaptive Lasso technique was used [30] on small samples [31] and the plasma DHA/ARA ratio (standardized coefficient = 61.15; 95%CI, 7.544 to 114.8; *p* = 0.0254) was selected for the total SRS score and the MDA-LDL levels (standardized coefficient = 5.63, 95% CI, −25.72 to 36.87; *p* = 0.723) (Table 3). For the plasma MDA-LDL levels, plasma SOD levels (standardized coefficient = 40.66, 95%CI, −60.13 to 21.19, *p* < 0.001) were selected (Table 4).

Collectively, these two statistical analysis results indicated that the plasma DHA/AR ratio is significantly associated with total SRS scores and plasma MDA-LDL levels.

### 2.7. Coefficients of Variation

The mean CVs for plasma PUFAs in the ASD and control groups were 0.03 (3.0%) and 0.096 (9.6%), respectively.

## 3. Discussion

The two measures were employed to overcome the limitations of the data interpretation in this small sample study: the adaptive Lasso technique was used to select appropriate covariates to maintain statistical efficiency [34], and data reliability was evaluated using coefficients of variation (SD/mean values, CV, %) in a small sample size [35]. In the present study, the mean CV for plasma PUFA was 0.03 (3.0%) in the 18 individuals with ASD and 0.096 (9.6%) in the control group. According to former studies, the mean CVs of the pharmacokinetics and tolerability of an oral evening dose of HLD 200 (54 mg), which is a psychostimulant used for the treatment of attention-deficit/hyperactivity disorder (ADHD), in 18 subjects with ADHD and 11 healthy adults was 7.8–17.7% [36] (Child), and that the reliability of a novel procedure in monitoring the flexibility of lower-limb muscle groups, indicating 8.3% for one muscle and 3.3–5% for another muscle in 10 athletes [35]. Additionally, liquid chromatographic-electrospray ionization mass spectrometric quantitative analysis of determination of volumes of buprenorphine, norbuprenorphine, nordiazepam and oxazepam in rat plasma reported ranging from 1.8 to 14.3% [37]. Taking these previously reported findings into account, the CV in the present study was appropriate.

With respect to the gender difference, there were no significant differences in the plasma variables and the total scores of SRS in the present study. However, plasma DHA levels tended to increase in 12 male individuals as compared with those in female individuals. The introduction of DHA into the diet produced sex-specific interactions on the fatty acid and produced a significant effect on the microbial profiles in males but not in females [38]. In general, plasma DHA levels are higher in female than in male subjects [39]. Although the tendency of higher plasma DHA levels is not significant in the present study, these were not of sufficient magnitude to warrant a move away from population-level diet recommendations for omega-3 PUFA [40]. With respect to the sex differences in social and behavioral symptoms in ASD, autistic girls employed more words than autistic boys and produced longer speech segments, whereas autistic boys spoke more slowly than control children and autistic girls did not differ from children without autism. Autistic boys interrupted others’ speech less often and produced longer between-turn pauses in children and adolescents aged 6–15 years [41]. The sex difference may be due to prenatal testosterone exposure [42]. Although there are different norms for boys and girls with ASD on several major screening tests, the algorithm of the ADI-R has not been reformulated [43]. Overall, no significant difference in the total SRS scores in this study may be due to the small sample size and less to do with the algorithm of the SRS.

Previous studies have suggested that lipid peroxidation is induced by downregulating endogenous antioxidant defense [44]. Examining confocal microscopy images has shown significantly higher levels of neuronal lipid peroxidation products and MDA in idiopathic ASD patients [45]. Previous animal studies indicated that DHA attenuates lipid peroxidation [46] and reduced lipid peroxidation because of lipid attenuation [47]. In the present study, a dietary assessment revealed no significant differences in dietary intake between the ASD and control groups. Therefore, the endogenous antioxidant capacity may be vulnerable, inducing lipid peroxidation.

The peroxidation of PUFAs has emerged as a key driver of oxidative damage to cellular membranes and leads to ferroptotic cell death [7]. FA oxidation is necessary for lipid-peroxidation-related ferroptosis [48], elevated NOX4, a major source of ROS, increases ferroptosis-dependent cytotoxicity by activating oxidative-stress-induced lipid peroxidation [49], which is caused by dysfunctional antioxidant systems [48]. Importantly, lipid-peroxidation-related ferroptosis produces increased MDA levels and decreased SOD levels via oxidant damage [24] (Mu) further induces iron metabolism (for example, through the transferrin receptor) with intense oxidative stress and inflammation [24]. Interestingly, the transferrin receptor protein levels were found to be enhanced by the ferroptosis activator erastin in wild-type cells [49]. Many previous studies supported our present findings indicating lipid-peroxidation-related ferroptosis related increased MDA levels and decreased SOD levels

A recent clinical study investigated the physiological effect of deep pressure, using an autism hug machine portable seat (AHMPS) on 20 children with ASD (mean 10.9 ± 2.26 years) [50]. These 20 individuals were divided into two groups. Group I used the AHMPS inflatable wraps model, and another group (group II) used the AHMPS manual pull model. Their heart rates were significantly decreased from the baseline in group 1; however, no change in heart rate variability was found in group II (*p* = 0.111) [50]. The Skin conductance was measured using galvanic skin response, which captured a significant decrease in group I (*p* < 0.0001), but no significant decrease was recorded in group II (*p* = 0.062) [50]. Since the indicator of skin conductance was closest to capturing autonomic alteration, the remaining effect of deep pressure was better in group I (*p* = 0.003) than in group II (*p* = 0.773) [50]. The AHMPS inflatable wrap model was more effective in decreasing physiological arousal [50] (Alif)eference; behavior problems may lead to academic and social disruptions in ASD, which led to the introduction of stability balls as an alternative seating method for children [51]. Therefore, the AHMPS inflatable wraps model may improve anxiety in those with ASD. However, there are few studies on the role of ferroptosis in the pathophysiology of ASD. Only one clinical study indicated that higher levels of ferroptosis scores were associated with immune activation in 293 ASD patients [52]. DHA plays an important role in lipid mediator production [53]. ARA is a major target of free radical attacks, which induce lipid peroxidation related to arachidonic acid hydroperoxide concentrations [54]. It is well known that a balanced dietary omega-6:omega-3 ratio is essential for brain development, impaired neocortical neurogenesis in the offspring [55]. It is important to note that the present findings indicate that the imbalance between increased DHA levels and decreased ARA levels in plasma may induce ferroptosis and contribute to the pathophysiology of ASD. The present study firstly elucidated that the important role of the imbalance between DHA and ARA in plasma may contribute to the pathophysiology of ASD.

The present findings indicate that the DHA/ARA ratio was 0.57. For reference, an ARA/DHA ratio greater than 1:1 is associated with improved cognitive outcomes [56], and a higher omega-6/omega-3 PUFA ratio improves antioxidant capacity [57]. Such unbeneficial balance may induce neurocognitive disorders [58,59]. Importantly, the imbalance between the omega-3 and increased omega-6 PUFAs are determinative for the consequences of oxidation and inflammation [60] and ferroptosis [61]. Therefore, the imbalance between the DHA levels and ARA levels in plasma may induce ferroptosis and contribute to the pathophysiology of ASD. Importantly, such imbalances between omega-3 and omega-6 PUFAs are determinative in neurocognitive disorders [62], and an increase in omega-6 PUFAs induce peroxidation and ferroptosis [63]. Therefore, an unbeneficial DHA/ARA ratio of 0.57 due to higher plasma ARA levels may reflect an imbalance between omega-3 and omega-6 PUFAs in plasma may induce ferroptosis, resulting in important mechanisms causing neurocognitive disorders, such as ASD. Importantly, the present study firstly shows that an unbeneficial DHA/ARA ratio contributes to inducing ferroptosis via lipid peroxidation in ASD.

Upon exceeding the buffering capacity of triglyceride storage in lipid droplets, omega-3 and omega-6 PUFA peroxidation leads to cytotoxic effects, such as ferroptosis [62]. Omega-6 PUFAs have greater susceptibility to lipid peroxidation and induce ferroptosis [63]. While, omega-3 PUFAs promote intracellular antioxidants synthesis and reduce the formation of hydroperoxides, inducing ferroptosis [63]. These findings suggested that an imbalance between DHA and ARA in plasma may induce ferroptosis as shown in the present study. Therefore, the DHA/ARA balance in the present study is important in ferroptosis-related lipid peroxidation.

Clinical evidence suggests that an ARA/DHA ratio greater than 1:1 is associated with improved cognitive outcomes [56]. In the present study, the DHA/ARA ratio may have resulted from decreased plasma DHA levels compared with plasma ARA levels, resulting in a less potent antioxidant capacity. Therefore, the present findings firstly note that this lower DHA/ARA ratio (less than 1.0) may be related to lower antioxidant capacity in DHA, inducing the lipid peroxidation product MDA-LDL and oxidative-stress-induced ferroptosis.

The plasma level of AdA (a member of the n-6 PUFA family) in the ASD group was significantly lower compared with the control group. AdA could play a role in resolving inflammation in vivo [64]. AdA is inflammation enhancer [65], and neuroinflammation has been suggested as being part of the pathophysiology of ASD [66]. AdA could contribute to neuroinflammation-related ASD. Plasma AdA levels in the ASD group were low in this study and thus may have contributed to ASD-related neuroinflammation.

The present study has limitations. First, the most prominent products were 4-hydroxyhexenal (4-HHE) from DHA and 4-hydroxynonenal (4-HNE) from ARA [53]. Of note, MDA-LDL appears to be the most mutagenic final product of lipid peroxidation, whereas 4-HHE and 4-HNE are considered bioactive markers of lipid peroxidation [48]. However, MDA-LDL is well known as a biomarker of lipid peroxidation in omega-3 and omega-6 PUFAs [1]. Therefore, we used plasma MDA-LDL as a biomarker of lipid peroxidation in this study.

Second, a small sample size increases the likelihood of a false null hypothesis (type II and type I errors), skewing the results and reducing the usefulness of the study. Adaptive Lasso is very good in terms of variable selection, estimation accuracy and high efficiency when small sample sizes are used [29,30]. Thus, using adaptive Lasso for statistical analyses may improve the reproducibility and sensitivity of the findings and reduce the likelihood of type II and type I errors. The present findings suggest a significant correlation between increased plasma MDA-LDL and decreased plasma SOD in ASD.

A previous review article reported similar results [15,67] as well as ferroptosis via lipid peroxidation [7,9]. The importance of plasma DHA and ARA ratios in social cognition [68] and neurodevelopment [69] is consistent with previous studies. Moreover, the DHA/ARA ratio plays an important role in cognitive learning [69]. Third, the ratio of case-to-control subjects in the present study was small: 2.5:1, potentially an unsuitable ratio. However, a case–control study of sleep apnea reported that 110 patients with idiopathic leg edema showed a higher body mass index than 55 controls (*p* = 0.03), suggesting that idiopathic bilateral leg edema is strongly associated with sleep apnea [70]. Additionally, deviated imbalance in the cases/controls was found in a recent case–control study including a case of 50 patients and 38 controls (Case/control = 2.1:5) [71]. Therefore, the ratio of our study (2.2:1) may be reasonable.

## 4. Materials and Methods

### 4.1. Subjects

Eighteen individuals with mild-to-moderate ASD and eight age-matched and healthy male and female control individuals (a total of 26 individuals) were randomly included in this study. Such an imbalance of case/controls was recognized in a lot of case–control studies: A case versus control study in relationship between the Shape of the spine and the width of Linea alba in children aged 6–9 years included 37 patients with the Linea alba and the 24 control subjects [72].

The information clearly and accurately represents the research, and the random recruitment process was handled ethically according to the IRB Submission Requirements. All 18 individuals were inducted in order of medical examination at our medical clinic (Fujimoto Clinic, Kobe City, Japan), taking place between January 2020 and June 2021. Eight healthy individuals in the same region as the ASD subjects were included in this study in order of application at our medical clinic during the same period.

Each of the 18 individuals with ASD received an independent clinical diagnosis of ASD. The 18 patients with ASD included 12 males and 6 females (mean age: 10.9 ± 5.6 years). The eight healthy controls included five males and three females (mean age: 9.6 ± 4.0 years) (Table 1). The diagnosis of ASD was made based on the Diagnostic and Statistical Manual of Mental Disorders, Fifth Edition (DSM-5). The Autism Diagnostic Interview, Revised (ADI-R) was used to confirm ASD diagnoses. The ADI-R was conducted by one of the authors (K.Y.), who is an expert in diagnosing ASD using the Japanese version of the ADI-R. The ADI-R is a semi-structured interview that is conducted with a parent. The 18 individuals with mild-to-moderate ASD had the core symptoms of the DSM-5 diagnostic criteria for ASD, without any abnormal neurological symptoms. The 18 individuals with mild-to-moderate ASD and the eight healthy controls were matched with respect to feeding habits, age and full-scale intelligent quotient (Table 1). No participants had any abnormalities in the results of their physical examinations or laboratory findings. Their intelligence quotients were assessed using the Wechsler Intelligence Scale for Children and Adolescents of 6–16 years of age (WISC-V) or the respective scale for adults (Wechsler Adult Intelligence Scale (WAIC-R). We used the Wechsler Intelligence Scale for Children, Third Edition (WISC-III) to include subjects with normal cognitive functions and a full-scale IQ of <70 [73]. We excluded comorbid psychiatric illnesses using the Structured Clinical Interview for DSM-5. Individuals were excluded from the study if they had epileptic seizures or obsessive-compulsive disorder or were diagnosed with any additional psychiatric or neurological conditions. All individuals with ASD were drug-naïve, and no individuals with ASD were taking dietary supplements. The healthy control subjects were recruited locally using an advertisement. All control individuals underwent a comprehensive assessment of their medical histories to exclude individuals with neurological or other medical disorders.

### 4.2. Precautions for Mitigating the Effects of Small Sample Size

As the small sample size in this study may limit the interpretation of the results, we employed three precautions. First, selecting the most appropriate method is needed with a small sample size [31]. A modified least absolute shrinkage and selection operator (adaptive Lasso) technique is useful for selecting appropriate covariates to account for confounding bias and thereby maintaining statistical efficiency [31]. Therefore, we used adaptive Lasso. Second, standard deviations (SDs) are useful for expressing variability. Therefore, the data reliability was evaluated using the coefficient of variation (CV, %), which is defined as the SD/the mean value to measure the relative variation of a random variable [74]. The CV was used to determine between- and within-subject reliability [74,75]. Third, to measure relative variation, the coefficient of variation (CV) is most often used [75].

### 4.3. Assessment of Social Behaviors

Social behaviors were assessed using the social responsiveness scale (SRS), which is used to distinguish ASD from other psychiatric disorders. The SRS is a 65-item questionnaire completed by the parents of subjects to quantitatively assess autistic traits to distinguish ASD from other psychiatric conditions [76], and it is used to assess the severity of autism symptoms [77].

### 4.4. Controlling for Dietary Intake and Assessment of Nutrient Intake

The PUFA composition of blood reflects dietary intake [78]. In this study, all 36 participants received the “Japanese Food Guide” (Ministry of Health, Labour and Welfare, and Ministry of Agriculture, Forestry and Fishers, Japanese Food Guide, 2012), which outlined the daily intake guide of nutrients and food, based on the “Overview of Dietary Reference Intake for Japanese (2010)” (Ministry of Health, Labour, and Welfare, 2010). All subjects were provided a diet based on the sample diet meal plan and menu (KAWASAKI FOODMODEL) (http://item.rakuten.co.jp/foodmodel/751741/, accessed on 10 September 2023), which was edited according to the “Japanese Food Guide” (Ministry of Health, Labour and Welfare, and Ministry of Agriculture, Forestry and Fishers, Japanese Food Guide, 2012). Moreover, we assessed daily food and nutrient intake. A semi-constructive questionnaire (DHQ) was performed in Japanese using a junior high school version of DHQ15 (DHQ Support Center, http://www.ebnjapan.org/, accessed on 10 September 2023). DHQ15 consists of 72 questions on the frequency of intake of 150 food and beverage items and cooking methods. DHQ15 was administered for two weeks before this study on randomly selected subsamples of seven individuals with ASD and seven controls. The nutrient items and portion sizes in the questionnaire were derived primarily from the data in the Overview of Dietary Reference Intake for Japanese (Ministry of Health, Labour, and Welfare. Overview of dietary reference Intake for Japanese. 2015). The DHQ15 data sheets were checked by our psychiatrist (K.Y.). If inaccurate information was recognized, the psychiatrist (K.Y.) confirmed the data using phone or e-mail. The validity of the DHQ15 has been verified [79]. The estimated intake of nutrients was calculated using a dedicated program for the DHQ system (DHQ support center, Tokyo, Japan) [80].

### 4.5. Measurement of Plasma PUFA, Cp, SOD and Tf Levels

#### 4.5.1. Blood-Sampling Procedures

Whole-blood samples were collected via venipuncture into EDTA tubes after 3-h fasts and then placed on ice. Nine-to-twelve hours of fasting before triglyceride measurement is considered appropriate [81]. Non-fasting triglyceride levels may replace fasting levels when evaluating cardiovascular disease risk [82]. Thus, a fasting time of 3 h after breakfast is reasonable and was applied to the present study. The blood samples were frozen at −80 °C until the plasma levels of the variables were analyzed at a clinical laboratory (SRL Inc., Tokyo, Japan).

#### 4.5.2. Plasma Levels of PUFAs

Blood samples were drawn from the study participants after at least 12 h of fasting. The serum specimens were separated, frozen and stored at −80 °C until use. The plasma fatty acid levels of the samples were measured using the gas chromatography method (SRL, Tokyo). Twenty-four polyunsaturated fatty acid fractionations, including EPA, DHA and AA concentrations, were measured. The intra- and inter-assay coefficients of ARA were 110.14 μg/mL (standard deviation (SD), 3.87; coefficient of variation (CV), 5.28%) and 100.63 μg/mL (SD, 5.51; CV, 5.48%), respectively, while those of DHA were 73.87 μg/mL (SD, 2.30; CV, 3.11%) and 68.07 μg/mL (SD, 2.30; CV, 3.33%). The plasma levels were expressed as the mean ± SD weight (%) of the total PUFAs.

#### 4.5.3. Plasma Levels of SOD

The intracellular SOD and cytochrome C molecules are specific and sensitive detection of the mitochondrial apoptotic signaling pathway. Both these two types may be highly amplified fluorescence for sensitive for detection of SOD in plasma [83]. Therefore, plasma SOD was estimated with the cytochrome c method using a SOD Assay Kit (Takara Bio Inc., Kusatsu, Japan) at a clinical analytical laboratory (SRL Inc., Tokyo, Japan). The assay sensitivity was 0.3 U/mL. The intra- and inter-assay coefficients were 2.11 and 2.10 U/mL, respectively.

#### 4.5.4. Plasma Levels of CP

To estimate plasma CP levels, a Bering BN II Nephelometer (Siemens Healthcare Diagnostics K.K., Malvern, PA, USA) was used. The assay sensitivity was 3.0 mg/dL.

#### 4.5.5. Plasma Levels of TF

A standard turbidimetric assay and an automated biochemical analyzer (JCA-BM8000 series, JEOL Ltd., Tokyo, Japan) were used to estimate plasma TF levels.

### 4.6. Plasma Levels of MDA-LDL

An enzyme-linked immunosorbent assay (ELISA) was used to measure the plasma levels of MDA-LDL; the measurement was performed at a clinical analytical laboratory (SRL Inc., Tokyo, Japan). The detection limit was 6.3 U/L, and the intra- and inter-assay coefficients were <5.6% and <9.4%, respectively [84].

### 4.7. Sex Differences in Plasma Variables and the Total SRS Scores

As shown in Table 5, there were no significant differences in plasma variables and total SRS scores.

### 4.8. Statistical Analyses

The relationship between plasma variables and SRS scores in the two groups was confirmed via multiple linear regression analysis (Table 2). To identify the most effective variables to interpret a small amount of sample data, adaptive Lasso was used. This statistical approach can correctly select variables in the presence of spurious covariates, that is, covariates unrelated to the outcome or exposure. The outcome-adaptive Lasso selects the propensity score model that includes all true confounders and predictors of the outcome [30,31] (Wah, bd). Adaptive Lasso is useful for consistent variable selection and identifying important variables [30] in small samples [31]. All of the statistical analyses were performed using SPSS version 27.0.

## 5. Conclusions

The present findings indicate an association between an increased plasma DHA/ARA ratio and increased plasma MDA-LDL levels, which may reduce plasma SOD levels. The correlation between increased MDA-LDL and decreased SOD in plasma, but not the insignificant contribution of TF to lipid peroxidation, may be caused by ferroptosis related to oxidative damage. A nonoptimal DHA/ARA ratio of 0.57 may promote ferroptosis via lipid peroxidation via the actions of the elongation of very long-chain fatty acid protein 5 (ELOVL5) and fatty acid desaturase 1 (FADS1). This neurobiological phenomenon may induce lipid peroxidation, inducing neuronal deficits and thus autistic social behaviors in individuals with ASD.

## 6. Perspective

Ferroptosis may be related to various neurodevelopmental disorders. However, most findings have been reported from experimental animal or cell studies, and thus more clinical approaches are necessary because many studies on cellular function show that signaling molecules also trigger ferroptosis. Most studies in this field appear to be conducted using clinical blood-sampling and biochemical methods. Clinical studies on ferroptosis may contribute to useful and extensive therapeutic alternatives for patients with neuronal diseases, including ASD. Moreover, studies on the molecular mechanisms underlying ferroptosis-induced cell death in neuronal functions are urgently needed.

## Figures and Tables

**Figure 1 ijms-24-14796-f001:**
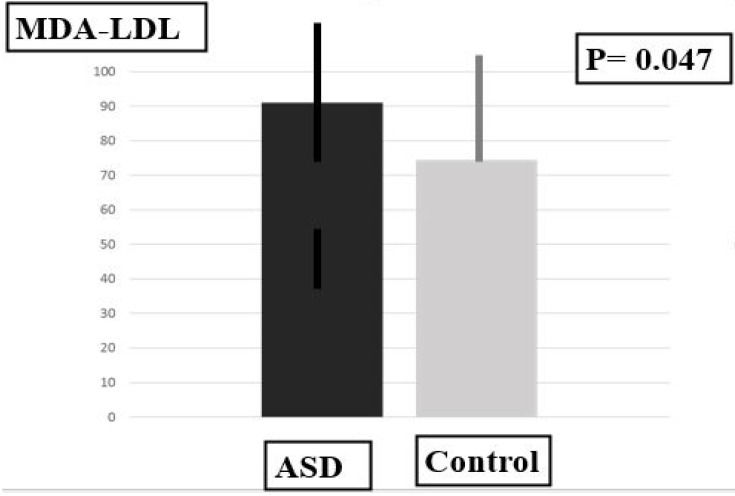
Plasma levels of MDA-LDL.

**Figure 2 ijms-24-14796-f002:**
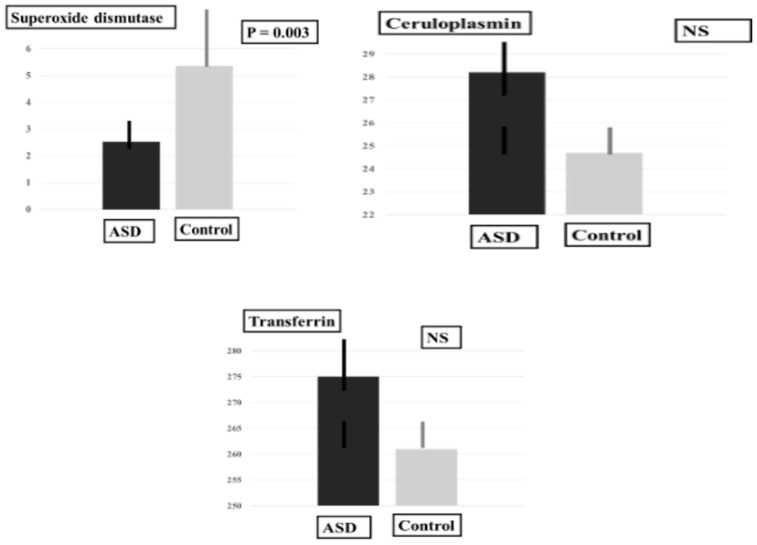
Plasma levels of antioxidant proteins.

**Figure 3 ijms-24-14796-f003:**
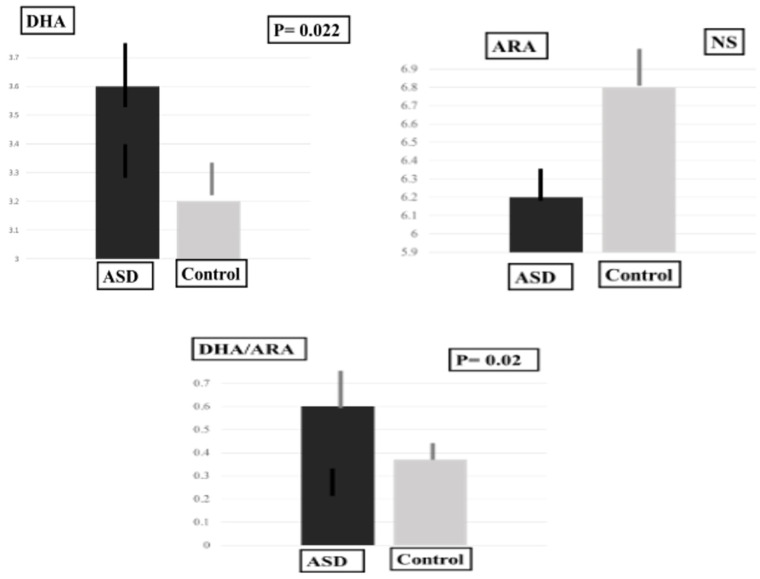
Plasma levels of DHA, ARA and DHA/ARA ratio.

**Table 1 ijms-24-14796-t001:** Subjects’ characteristics and plasma levels of antioxidants and plasma variables and the ABC total scores.

Variables		ASD (*n* = 18)	Control (*n* = 8)	U	*p* Values
	Age (Years)	10.9 ± 5.6	9.6 ± 4.0	63	0.64
	Sex (Male/Female)	12/6	5/3	χ^2^ = 0.4	0.84
Subject Characteristics	Domain A	24.4 ± 3.7	NA		
	Domain B (Communication)	12.0 ± 2.3	NA		
	Domain C (Stereotyped Behaviors)	10.8 ± 3.5	NA		
Antioxidant	Cp (mg·dL)	28.17 ± 7.02	24.63 ± 6.78	55.00	0.37
	Tf (mg/dL)	275.39 ± 41.89	261.25 ± 24.06	56.00	0.40 *
	SOD (U·mL)	2.53 ± 0.46	5.36 ± 4.39	12.50	0.003 *
Biomarkers	MDA-LDL	91.00 ± 16.70	74.50 ± 18.88	36.50	0.047 *
	DHA	3.57 ± 0.98	3.25 ± 0.84	31.50	0.022 *
	ARA	6.23 ± 1.08	6.79 ± 1.58	53.00	0.31
	DHA/ARA	0.60 ± 0.16	0.37 ± 0.07	10.50	0.02 *
	Adrenic Acid	0.26 ± 0.25	0.27 ± 0.06	32.00	0.03 *

Data are represented as mean ± SD (Mann-Whitney U test). * *p* < 0.05, versus normal controls. Cp, ceruloplasmin; Tf, transferrin; SOD, superoxide dismutase; MDA-LDL, malondial-dehyde modified low-density lipoprotein.

**Table 2 ijms-24-14796-t002:** Results of the multiple linear regression analysis.

Model		Model	Model		Coefficients	
		R^2^	*p*-Value	B	Beta	*p* Value
					Coefficients	
DHA		0.991	0.000 **			
	α-linolenic acid			0.535 ± 0.155	0.107	0.013 *
	DPA			0.498 ± 0.193	0.088	0.04 *
	GLA			−1.166 ± 0.306	−0.157	0.007 *
	SOD			−0.06 ± 0.015	−0.170	0.03 *
	SRS total			0.000009 ± 0.001	0.092	0.057
	Group (1 = ASD, 2 = control)			−0.057 ± 0.102	−0.26	0.591
DHA/ARA		0.972	0.000 **			
	ARA			−0.057 ± 0.102	−0.541	0.01 *
	α-linolenic acid			−0.267 ± 0.102	−0.316	0.04 *
	Adrenic acid			−0.070 ± 0.147	−0.037	0.650
	SOD			0.016 ± 0.015	0.242	0.326
	SRS total score			0.000009 ± 0.001	−0.023	0.901
	Group (1 = ASD, 2 = control)			0.020 ± 0.057	0.05	0.731

* *p* < 0.05, ** *p* < 0.01, R^2^, R-squared values; B, Unstandardized coefficients; SOD, Superoxide dismutase; SRS, Social Responsiveness Scale.

**Table 3 ijms-24-14796-t003:** Results of adaptive Lasso.

Variables	Standardized Coefficients	SE	95% CL		*p* Value
			Lower Bound	Upper Bound	
SRS total scores					
DHA/ARA	61.155	27.353	7.546	114.776	0.125
MDA-LDL					
DHA/ARA	5.633	15.91	−25.572	36.837	0.712
SOD	−40.060	9.935	−0.135	−31.186	<0.001

MDA-LDL, malondialdehyde-modified low-density lipoprotein; SRS, Social Responsiveness Scale.

**Table 4 ijms-24-14796-t004:** The intake of nutrients in the random subsamples of 7 of the 17 individuals and 5 of the 7 normal controls.

		ASD	Control	U	*p* Value
		(*n* = 7)	(*n* = 5)		
Age (years)		11.4 ± 4.3	11.4 ± 3.2	14.0	0.87
Fat (g/day)		72.2 ± 30.1	87.4 ± 25.8	12.0	0.43
Unsaturated		14.8 ± 4.4	18.7 ± 5.5	9.0	0.20
Fatty acid (g/day)					
PUFAs	Omega-3 PUFAs (g/day)	2.6 ± 0.8	3.1 ± 0.5	9.0	0.20
	Omega-6 PUFAs (g/day)	12.1 ± 3.9	15.9 ± 5.1	11.0	0.34
	EPA (mg/day)	181.2 ± 118.7	176.2 ± 73.6	15.5	0.76
	DHA (mg/day)	332.6 ± 170.4	345.0 ± 83.31	15.5	0.76
	ARA (mg/day)	168.1 ± 17.1	221.0 ± 87.7	11.5	0.34
Protein (g/day)		78.1 ± 25.8	89.2 ± 25.8	13.0	0.53
Animal protein (mg/day)		32.0 ± 9.1	30.4 ± 14.7	14.0	0.64
Cholesterol (mg/day)		139.1± 186.4	31.9 ± 10.3	17.0	1.00
Carbohydrates (g/day)		286.2 ± 62.1	304.21 ± 72.4	14.0	0.64
Copper (mg/day)		1.0 ± 0.2	1.3 ± 0.5	7.0	0.56
Iron (mg·day)		7.4 ±1.6	9.2 ± 3.5	6.0	0.41

PUFA, polyunsaturated fatty acid; EPA, eicosapentaenoic acid; DHA, docosahexaenoic Acid; ARA, arachidonic acid. Values are mean ± SD.

**Table 5 ijms-24-14796-t005:** Sex differences in plasma variables and the total SRS scores.

	MDA-LDL	DHA	ARA	DHA/ARA	Adrenic	SOD	Total SRS
					Acid		Scores
Male (*n* = 12)	97.17 ± 17.06	3.85 ±1.07	6.16 ± 0.65	0.62 ± 0.16	0.27 ± 0.30	2.59 ± 0.42	89.03 ± 30.18
Female (*n* = 6)	88.67 ±17.36	3.00 ± 0.60	6.37 ± 1.72	0.56 ± 0.17	0.24 ± 0.13	2.40 ± 0.56	70.50 ± 44.36
U	34.50	15.50	33.00	25.50	33.00	28.50	24.00
*p* values	0.89	0.053	0.82	0.34	0.82	0.49	0.29

MDA-LDL, malondialdehyde-modified low-density lipoprotein; SRS, Social Responsiveness Scale.

## Data Availability

The data presented in this study are available on request from the corresponding author. The data are not publicly available due to intellectual property concerns.

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
