# Peer review of "Lipid Peroxidation of the Docosahexaenoic Acid/Arachidonic Acid Ratio Relating to the Social Behaviors of Individuals with Autism Spectrum Disorder: The Relationship with Ferroptosis"

_ijms, 2023, doi:10.3390/ijms241914796_

Round 1
Reviewer 1 Report (Previous Reviewer 2)
This is a potentially interesting study. However, I have some concerns that should be addressed before being recommended for publishing. 1. What does the schematic entitled “DHA/Arachidonic Acid Ratio and Lipid Peroxidation in Autistic Social Behaviors” refers to? Is that a graphical summary of the manuscript?
2. Introduction: “Lipid peroxidation causes cell damage and various human health issues [1]” What kind of health problems? Pls provide some examples.
3. I am afraid that I do not understand the intended meaning of the following sentence “The intricate correlation between ferroptosis and ASD and provided a promising ferroptosis score predict the molecular clusters and immune infiltration cell profiles in children with ASD [9].”
4. I am afraid that I do not understand the intended meaning of the following sentence: “Importantly, the presence of ferroptosis by lipid peroxidation in this study was based on proven of increased plasma levels of MDA-LDL [10,11]”. The manuscript has numerous typos, grammatical and sentence construction errors that make it, at times, difficult to understand.
5. Introduction is not easy to follow and would benefit from restructuring.
6. I don’t understand this goal: “(a) how the effects of plasma PUFAs induce ferroptosis related to autistic social behaviors”. What do the authors mean by the “effects of plasma PUFA”, and what is the “ferroptosis related to autistic social behaviors” (where is the link between the two)?
7. “which variable of PUFA and MDA-LDL mainly contributes to the development of autistic social symptoms” – what do the authors mean by the “variable”?
8. I do not understand the purpose of the following paragraph at the end of the Introduction- “Methodological advancements have resulted in improved unbiased estimation from observational data. The outcome-adaptive lasso ……….”
9. In the methods section – the authors tried to justify why they have 18 cases and only 8 controls by stating this is seen in many studies. However, then they wen on to cite only ine and about porphyria. I am afraid this is not an appropriate example for comparison with ASD.
10. Not just that the sample size is small, and that there is two times less controls then cases, the sample also include both males and females – making it actually impossible to make any conclusions, especially not about the role of sex. Furthermore, no power calculations for this type of study are provided.
11. “In this study, all 36 participants….” 18+8 = 26
12. “Blood samples were drawn from the study participants after at least 12 h of fasting.” – this actually contradicts to information provided in chapter 4.4.1.
13. In the results section, the authors say “The total SRS score of 29 young men (aged 7–21 years old) with moderate-to-severe ASD was 120.21 [32]”. Which 29 men? The sample is 18 cases and 8 controls (stated in the methods section)?
14. This statement “the differences in the core symptoms for each sex are inconsistent [33]” is actually not correct and there are numerous references supporting that.
15. Please provide exact p values not p = 0.000.
16. He Discussion, provides detailed reviews of some papers, instead of actually putting this study into the context of relevant literature. I ama afraid that, similar to the introduction, the Discussion should benefit from restructuring and a scholarly discussion of the present findings.
17. The conclusion is actually overstated (data are overinterpreted).
I am afraid that there are numerous grammatical etc errors in the manuscript, that make some statements/sentences very difficult to understand/follow.
Author Response
Please see the attachment.

Reviewer 2 Report (New Reviewer)
Dear Authors, I have read the manuscript titled "Lipid Peroxidation of the Docosahexaenoic acid/Arachidonic acid Ratio relating to the Social Behaviors of Individuals with Autism Spectrum Disorder: the Relationship with Ferroptosis" The original paper based on a clinical study, focuses on studying the relationship between ferroptosis induced by lipid oxidation and ASD.
The manuscript complies with the requirements of the journal, the bibliographic references in large numbers are appropriate for the presented topic, most of them being quite recent. I appreciate that you also gave us the Conclusions section. The 5 tables and the scheme from the abstract inserted in the manuscript improve the reading and make it easier to understand some aspects.
However, I have a number of questions and suggestions:
1. Throughout the material, there are countless places marked in yellow. Was it an omission in the submission or what does this mean?
2. The author numbering seems to be wrong. There are 3 authors and the last author received no. 4. Does the Department of Pediatrics receive a number? Please correct these issues.
3. You used several types of fonts in technical editing. Compare, for example, the bibliography with the rest of the text. Likewise within the 5 tables, for example, table no. 3.
4. To determine the enzymatic activity of superoxide dismutase you used the cytochrome C method. Could the authors justify why they chose this method? We know that other types of methods can be used to determine SOD. What was the reasoning behind choosing your method?
5. The results obtained could be represented in the form of graphs. I recommend inserting such graphics.
6. As a perspective you stated that studies on molecular mechanisms are needed. Do the authors already have a possible study underway?
7. The English language could be improved.
The English language could be improved.
Author Response
Please see the attachemnt.

This manuscript is a resubmission of an earlier submission. The following is a list of the peer review reports and author responses from that submission.
Round 1
Reviewer 1 Report
The authors could improve the clarity of their writing by providing more detailed explanations of their proposed method. This would help readers better understand the methodology and how it differs from other works in the same domain. Additionally, the authors should discuss the well-motivated differences with the proposed work to highlight its contribution. Furthermore, the authors should provide a more comprehensive bibliography of the materials and methods used in their study. This would allow readers to understand better the research context and the latest models used as fair baselines. The authors should also focus on improving the discussion section of their paper. They should provide a detailed discussion of the results and the implications of their findings for individuals with Autism Spectrum Disorder. Additionally, they should describe future improvement methods to encourage further research in this area. By following these suggestions, the authors can improve the effectiveness of their writing and make a more significant contribution to the field.
Extensive editing of the English language is required.
Author Response
Reviewer 1’s Comments:
The authors could improve the clarity of their writing by providing more detailed explanations of their proposed method.
Our answer
We have written by providing more detailed explanations of their proposed method on Section 4 SUBJECTS AND METHODS by marked with yellow colors follows:
- SUBJECTS AND METHODS
- Subjects
Eighteen individuals with mild-to-moderate ASD and eight age-matched and healthy male and female control individuals were randomly included in this study. The information clearly and accurately represents the research, and the random recruitment process was handled ethically according to IRB Submission Requirements. All 18 individuals were inducted in order of medical examination at our medical clinic (Fujimoto Clinic, Kobe City in Japan), taking place between January 2020 and June 2021. Eight healthy individuals in the same region as the ASD subjects were included in this study in order of application at our medical clinic during the same period.
The authors should discuss the well-motivated differences with the proposed work to highlight its contribution
Our answer
We have discussed the well-motivated differences with the proposed work to highlight its contribution on the duscussion section on page 10 to 11 as follows: “The peroxidation of PUFAs has emerged as a key driver of oxidative damage to cellular membranes and leads to ferroptotic cell death [36] PUFA oxidation is necessary for lipid-peroxidation-related ferroptosis [37](Yang). Of note, elevated NOX4, a major source of ROS, increases ferroptosis-dependent cytotoxicity by activating oxidative-stress-induced lipid peroxidation [38] which is caused by dysfunctional antioxidant systems [39]. Importantly, [19]. Interestingly, the transferrin receptor protein levels were found to be enhanced by the ferroptosis activator erastin in wild-type cells [40].
Regarding the treatment of ASD, a recent study reported that [41]. However, . Only one clinical study indicated that higher levels of ferroptosis scores were associated with immune activation in 293 ASD patients [42]
The present findings indicate that such imbalances between omega-3 and omega-6 PUFAs are determinative in inducing oxidative stress [52].. Oxidative stress-induced lipid peroxidation eventually promotes to ferroptosis [50].. Therefore, the unbeneficial DHA/ARA ratio in the present study may induce ferroptosis via lipid peroxidation, resulting in neuronal deficiency in the pathophysiology of ASD. Additionally, the lower DHA/ARA ratio (less than 1.0) in the present study may be related to less antioxidant capacity in DHA, inducing the lipid peroxidation product MDA-LDL and oxidative stress induced ferroptosis.ï¼’
The authors should provide a more comprehensive bibliography of the materials and methods used in their study. This would allow readers to understand better the research context and the latest models used as fair baselines.
Our answer
We provided a more comprehensive bibliography of the materials and methods used in their study marked with yellow color on the 4. SUBJECTS AND METHODS, 4.1. Subjects, 4.4. Controlling for dietary intake and assessment of nutrient intake, and 4.7. Statistical analyse on page
They should provide a detailed discussion of the results and the implications of their findings for individuals with Autism Spectrum Disorder
Our answer
We have provide a detailed discussion of the results and the implications of their findings for individuals with Autism Spectrum Disorder on page 11, marker with yellow color as follows: “Regarding the treatment of ASD, a recent study reported that [41]. However, . Only one clinical study indicated that higher levels of ferroptosis scores were associated with immune activation in 293 ASD patients [42]
They should describe future improvement methods to encourage further research in this area.
Our answer
We have described future improvement methods to encourage further research in this area on page 19, 6. Perspectives as follows:
“5. Perspective
Ferroptosis may be related to various neurodevelopmental disorders. However, most findings have been reported from experimental animal or cell studies, and thus, more clinical approaches are necessary because many studies on cellular function show that signaling molecules also trigger ferroptosis. Most studies in this field appear to be conducted using clinical blood-sampling and biochemical methods. Clinical studies on ferroptosis may contribute to useful and extensive therapeutic alternatives for patients with neuronal diseases, including ASD. Moreover, studies on the molecular mechanisms underlying ferroptosis-induced cell death in neuronal functions are urgently needed.”
Reviewer 2 Report
In the introduction, the entire first paragraph is actually meaningless, and the rest of the manuscript is very similar, with so many typos, grammatical, sentence construction errors etc that it is almost impossible to read it. Even in the Conclusion, the authors write "The correlation of increased MDA-LDL and decreased SOD in plasma but not no significant contribution of Tf to lipid peroxidation may be induced via ferroptosis in related to oxidative damage." - this sentence, I am afraid is very difficult, if not impossible to understand. In the introduction, the entire first paragraph is actually meaningless, and the rest of the manuscript is very similar, with so many typos, grammatical, sentence construction errors etc that it is almost impossible to read it. Even in the Conclusion, the authors write "The correlation of increased MDA-LDL and decreased SOD in plasma but not no significant contribution of Tf to lipid peroxidation may be induced via ferroptosis in related to oxidative damage." - this sentence, I am afraid is very difficult, if not impossible to understand.Author Response
In the introduction, the entire first paragraph is actually meaningless, and the rest of the manuscript is very similar, with so many typos, grammatical, sentence construction errors etc that it is almost impossible to read it.
Our answer
We have changedthe sentences in the bfirst paragraph as fellow: Lipid peroxidation is a process under which oxidants attack lipids containing carbon–carbon double bonds and bonds such as polyunsaturated fatty acids (PUFAs). Lipid peroxidation causes cell damage and various human health [1]. On page 4.
Even in the Conclusion, the authors write "The correlation of increased MDA-LDL and decreased SOD in plasma but not no significant contribution of Tf to lipid peroxidation may be induced via ferroptosis in related to oxidative damage." - this sentence, I am afraid is very difficult, if not impossible to understand.
Our answer
We changed the sentences in the Conclusion section as follow: “A nonoptimal DHA/ARA ratio of 0.57 may promote ferroptosis via lipid peroxidation via the actions of elongation of very long-chain fatty acid protein 5 (ELOVL5) and fatty acid desaturase 1 (FADS1). This neurobiological phenomenon may induce lipid peroxidation, inducing neuronal deficits and thus autistic social behaviors in individuals with ASD.”
Reviewer 3 Report
Several comments given in the manuscript as follows.
1. Nothing truly unique in its current state. Because of the lack of a novel, the current submission looks to be a replication or modified work. The authors must describe their novel in detail. This work should be rejected owing to a major issue.
2. Additional figures in the introduction would improve the quality of the present article. Please provide it.
3. Autism spectrum disorder (ASD) is a developmental disorder that affects communication and behaviour. Please include this explanation along with supporting reference as follows, doi: 10.3390/bioengineering9020048
4. Section 2.2 Dietary nutrients, please give brief explanation and the basis of its information.
5. multiple linear regression in section 2.4 should given further explained to make it clearer.
6. What is urgency of Lasso analysis performed?
7. The authors need to explain more clearly the basis of patient selection. Has any protocol, basis, or standard been followed? The present form was not unclear since the patient involved is heterogeneous with a small number. It would direct impact the results that lead to inappropriate work. One of the major issues that need to be solved by the authors after the revision stage.
-
Author Response
Several comments given in the manuscript as follows.
- Nothing truly unique in its current state. Because of the lack of a novel, the current submission looks to be a replication or modified work. The authors must describe their novel in detail. This work should be rejected owing to a major issue.
Our answer
We have provided the novel findings related discission on page 4 as follows:
“PUFA-containing phospholipids (PUFA-PLs) are highly susceptible to lipid peroxidation under oxidative stress [5]. The imbalance of oxidation and anti-oxidation is regulated by numerous factors and pathways inside and outside the cell in relation to ferroptosis [6]. This easily induces ferroptosis, leading to membrane damage and cell death [7]. Ferroptosis is a form of regulated cell death dependent on iron and reactive oxygen species (ROS) and characterized by the accumulation of lipid peroxides [8].
Ferroptosis caused by lipid peroxidation is demonstrated by increased MDA-LDL plasma levels [9,10]. PUFAs play important roles in ferroptosis because of the formation of peroxyl radicals, leading to irreparable membrane damage and cell death [6]. The misregulation or depletion of PUFA-protective enzymes and molecules leads to ferroptotic damage in lipid peroxidation [7]. Ferroptosis is mediated by PUFA lipid peroxidation [11] via polyunsaturated phosphatidylethanolamines or complexes of 15-lipoxygenase (15LOX) and phosphatidylethanolamine-binding protein 1 (Lamade et al. 2022). DHA triggers ferroptosis by upregulating nuclear receptor coactivator 4 (NCOA4)[12]. Furthermore, arachidonic acid (ARA) facilitates ferroptosis by elongating very long-chain fatty acid protein 5 (ELOVL5) and fatty acid desaturase-1 (FADS1)[13] or arachidonic acid 15-lipoxygenase [14]. Therefore, it is important to note that ferroptosis-related cell death may be related to neuronal deficiency, resulting in a causal factor in ASD. However, there are few studies on the role of ferroptosis.”
- Additional figures in the introduction would improve the quality of the present article. Please provide it.
Our answer
We have provided the new figure as the graphical abstract,
- Autism spectrum disorder (ASD) is a developmental disorder that affects communication and behaviour. Please include this explanation along with supporting reference as follows: doi: 10.3390/bioengineering9020048
Our answer
We have introduced this reference and described related discussion as follows;” Regarding the treatment of ASD, a recent study reported that [41]. However, . Only one clinical study indicated that higher levels of ferroptosis scores were associated with immune activation in 293 ASD patients [42] on the Discussion section on page 11.
”
- Section 2.2 Dietary nutrients, please give brief explanation and the basis of its information.
Our answer
We have provided brief explanation and the basis of this information in the Subjects and method section on page 16 as fellows:
” 4.4. Controlling for dietary intake and assessment of nutrient intake
As plasma fatty acid levels may be confounded by prior dietary intake [65], all 36 participants received the “Japanese Food Guide” (Ministry of Health, Labor, and Welfare and Ministry of Agriculture, Forestry, and Fishers; Japanese Food Guide, 2012), which outlines the recommended daily intake of nutrients for Japanese citizens. To assess the daily nutrient intake, a semi-constructive questionnaire for Japanese people (DHQ15) was conducted (DHQ Support Center, http://www.ebnjapan.org/). DHQ15 consists of 72 questions on the frequency of food intake and was applied to all participants one month before the study. The validity of DHQ15 has been verified [66].
- Mutiple linear regression in section 2.4 should given further explained to make it clearer.
Our answer
We have given the results of the linera regression analysis in the Result section on page 8 as fellow;
- ” Results of the multiple linear regression analysis
A multiple linear regression analysis revealed that the plasma DHA level (R2 = 0.997, p = 0.000) and plasma DHA/ARA ratio (R2 = 0.972, p = 0.001) were significantly associated with adjustments in plasma variables and the total SRS scores in the two subject groups (Table 3). These findings revealed that the plasma DHA level and the plasma DHA/ARA ratio can predict all the variables of the two groups. The use of plasma alpha-linolenic acid levels as a dependent variable showed the significant contribution of the plasma DHA/ARA level (unstandardized coefficients, B= -0.267 ± 0.102, β = -0.316, p = 0.04) (Table 3). Therefore, the plasma DHA level and plasma DHA/ARA ratio distinguished the ASD group from the control group.”
- What is urgency of Lasso analysis performed?
Our answer
We have provided the urgency of adaptive Lasso in the Subjects and methodsection on page 15 as follows:”As the small sample size in this study may limit the interpretation of the results, we employed three precautions. First, selecting the most appropriate method is needed in s small sample size [61]. A modified least absolute shrinkage and selection operator (adaptive Lasso) technique is useful for selecting appropriate covariates to account for confounding bias and thereby maintain statistical efficiency [61]. Therefore, we used adaptive Lasso.”
- The authors need to explain more clearly the basis of patient selection. Has any protocol, basis, or standard been followed? The present form was not unclear since the patient involved is heterogeneous with a small number. It would direct impact the results that lead to inappropriate work. One of the major issues that need to be solved by the authors after the revision stage.
Our answer
We have provided the appropriate and right method for subject selection in the heterogenous small sample size in the Subjects and method section on page 15 as follows:
” 4. Subjects and methods
- Subjects
Eighteen individuals with mild-to-moderate ASD and eight age-matched and healthy male and female control individuals were randomly included in this study. The information clearly and accurately represents the research, and the random recruitment process was handled ethically according to IRB Submission Requirements. All 18 individuals were inducted in order of medical examination at our medical clinic (Fujimoto Clinic, Kobe City in Japan), taking place between January 2020 and June 2021. Eight healthy individuals in the same region as the ASD subjects were included in this study in order of application at our medical clinic during the same period.”
Further, 4.2. Precautions for mitigating the effects of small sample sizes
With respect to the two measures for small sample size, we have provided descriptions as fellows: “ 4.2. Precautions for mitigating the effects of small sample sizes
As the small sample size in this study may limit the interpretation of the results, we employed three precautions. First, selecting the most appropriate method is needed in s small sample size [59]. A modified least absolute shrinkage and selection operator (adaptive Lasso) technique is useful for selecting appropriate covariates to account for confounding bias and thereby maintain statistical efficiency [59]. Therefore, we used adaptive Lasso. Second, standard deviations (SDs) are useful for expressing variability. Therefore, the data reliability was evaluated using the coefficient of variation (CV, %), which is defined as the SD/the mean value [60] to measure the relative variation of a random variable. The CV was used to determine between- and within-subject reliability [62]. Third, to measure relative variation, the coefficient of variation (CV) is most often used [62].”
Round 2
Reviewer 1 Report
It is reassuring that the authors have considered and incorporated the suggested revisions into their paper. However, while the article is nearly ready for acceptance, a minor modification requires attention. Specifically, conducting an English spell-check to ensure accuracy would be helpful. Once the revision has been addressed, the paper should be deemed suitable for acceptance.
Moderate editing of the English language is required.
Author Response
Reviewer 1’s Comments:
Specifically, conducting an Englash spell-check to ensure accuracy would be helpful
Our answer
Thank you so much for your positive comments.
We have carefully checked English spellcheck three times
Reviewer 3 Report
Good job in the previous revision. Some comments given in this stage.
1. In the abstract section, quantitative data must be included.
2. As the last note in your abstract, please provide a "take-home" message.
3. Rearrange the keywords so that they are in alphabetical order.
4. I am encouraging the authors to not use abbreviations in the keywords.
5. Please explain in brief the correlation between ASD and anxiety, also refer the previous literature as follows: https://doi.org/10.3390/bioengineering9040157
-
Author Response
Reviewer 3’s Comments:

We are well pleased for you approval.
- In the abstract section, quantitative data must be included.
Our answer
We added important quantitative data in the Abstract marked with yellow color such as “Abstract: Polyunsaturated fatty acids (PUFAs) undergo lipid peroxidation and conversion into malondialdehyde (MDA). MDA reacts with acetaldehyde to form malondialdehyde-modified low-density lipoprotein (MDA-LDL). We studied unsettled issue in the association between MDA-LDL and pathophysiology of ASD in 18 individuals with ASD and 8 age-matched controls. Social behaviors were assessed using Social Responsiveness Scale (SRS). To overcoming the small samples problem, adaptive Lasso was used to enhance the interpretability accuracy, and coefficient of variation was used for variable selections. Plasma levels of MDA-LDL levels (91.00 ±16.70 vs 74.50 ± 18.88) and the DHA/arachidonic acid (ARA) ratio (0.60 ± 0.16 vs. 0.37 ± 0.07) were significantly higher and superoxide dismutase levels were significantly lower in the ASD group than those in the control group. Total SRS scores in the ASD group were significantly higher than those in the control group. Multiple linear regression analysis and adaptive Lasso revealed an association between an DHA/ARA ratio (0.57) with total SRS scores and increased MDA-LDL levels in plasma. This unbeneficial DHA/ARA ratio induce ferroptosis via lipid peroxidation, and mediated association between increased MDA-LDL and decreased SOD, resulting in neuronal deficiencies. This unbeneficial DHA/ARA ratio induced-ferroptosis contribute to autistic social behaviors, and became available for therapy.
- As the last note in your abstract, please provide a "take-home" message.
Our answer
We added a “take-home” messege in the last phrase in the Abstract marked with blue color as fwllow: This unbeneficial DHA/ARA ratio induced-ferroptosis contribute to autistic social behaviors, and became available for therapy.
- Rearrange the keywords so that they are in alphabetical order.
Our answer
We arranged the keywords in alphabetical order.
- I am encouraging the authors to not use abbreviations in the keywords.
Our answer
We did not use abbreviation in the keywords.
- Please explain in brief the correlation between ASD and anxiety, also refer the previous literature as follows: https://doi.org/10.3390/bioengineering9040157
Our answer
We have explained in brief the correlation between ASD and anxiety, also refer the previous literature as fellows: htttps://doi,org/10.3390/ bioengineering9040157 as fellows on page 11 marked with yellow color as follow: Regarding the treatment of ASD, according to a recent study, most individuals with ASD have sensory processing disorder, inducing inability to behavioral response to sensory input due to impaired sensory input [46]. A recent clinical study investigate the physiological effect of deep pressure, using an autism hug machine portable seat (AHMPS) in 20 children with ASD (mean 10.9 ± 2.26 years) [46]. These 20 individuals were divided into two groups. Group I used the AHMPS inflatable wraps model and another group (group II) used the AHMPS manual pull model. Heart rate was significantly decreased from the baseline in group 1, however, no change of heart rate variability was found in group II ( p = 0.111)[46]. Skin conductance was measured using galvanic skin response captured the significant decrease in group I (p <.0001), but no significant decrease was recorded in group II (p = 0.062) [46]. Since the indicator of skin conductance was closest in capturing autonomic alteration, the remaining effect of deep pressure was better in group I (p = 0.003) than in group II (p = 0.773)[46]. The AHMPS inflatable wrap model was more effective in decreasing physiological arousal [46]. Of reference, behavior problems may lead to academic and social disruptions in ASD led to the introduction of stability balls as an alternative seating method for children [47]. Therefore, the AHMPS inflatable wraps model may improve anxiety in ASD.
However, there are few studies on the role of ferroptosis in pathophysiology in ASD. Only one clinical study indicated that higher levels of ferroptosis scores were associated with immune activation in 293 ASD patients [48]. DHA plays an important role in lipid-mediator production [49]. ARA is a major target of free radical attacks, which induce lipid peroxidation related to arachidonic acid hydroperoxide concentrations [50]. Importantly, the present findings indicated that the imbalance between increased DHA levels and decreased ARA levels in plasma may induce ferroptosis via neuronal apoptosis [41,51], and contribute to the pathophysiology of ASD. The present firstly elucidated the important role of imbalance between DHA and ARA in plasma may contributed to pathophysiology of ASD.”
Round 3
Reviewer 3 Report
-
-